# Dynamic Regulation of Mitochondrial [Ca^2+^] in Hippocampal Neurons

**DOI:** 10.3390/ijms232012321

**Published:** 2022-10-14

**Authors:** Liliya Kushnireva, Kanishka Basnayake, David Holcman, Menahem Segal, Eduard Korkotian

**Affiliations:** 1Department of Brain Sciences, The Weizmann Institute of Science, Rehovot 7610001, Israel; 2Computational Biology and Applied Mathematics (IBENS), Ecole Normale Supérieure-PSL, 75005 Paris, France

**Keywords:** mitochondria, caffeine, CCCP, calcium surges, spikelets

## Abstract

While neuronal mitochondria have been studied extensively in their role in health and disease, the rules that govern calcium regulation in mitochondria remain somewhat vague. In the present study using cultured rat hippocampal neurons transfected with the mtRCaMP mitochondrial calcium sensor, we investigated the effects of cytosolic calcium surges on the dynamics of mitochondrial calcium ([Ca^2+^]m). Cytosolic calcium ([Ca^2+^]c) was measured using the high affinity sensor Fluo-2. We recorded two types of calcium events: local and global ones. Local events were limited to a small, 2–5 µm section of the dendrite, presumably caused by local synaptic activity, while global events were associated with network bursts and extended throughout the imaged dendrite. In both cases, cytosolic surges were followed by a delayed rise in [Ca^2+^]m. In global events, the rise lasted longer and was observed in all mitochondrial clusters. At the end of the descending part of the global event, [Ca^2+^]m was still high. Global events were accompanied by short and rather high [Ca^2+^]m surges which we called spikelets, and were present until the complete decay of the cytosolic event. In the case of local events, selective short-term responses were limited to the part of the mitochondrial cluster that was located directly in the center of [Ca^2+^]c activity, and faded quickly, while responses in the neighboring regions were rarely observed. Caffeine (which recruits ryanodine receptors to supply calcium to the mitochondria), and carbonyl cyanide m-chlorophenyl hydrazine (CCCP, a mitochondrial uncoupler) could affect [Ca^2+^]m in both global and local events. We constructed a computational model to simulate the fundamental role of mitochondria in restricting calcium signals within a narrow range under synapses, preventing diffusion into adjacent regions of the dendrite. Our results indicate that local cytoplasmic and mitochondrial calcium concentrations are highly correlated. This reflects a key role of signaling pathways that connect the postsynaptic membrane to local mitochondrial clusters.

## 1. Introduction

Mitochondria (MT) are the main energy provider of living organisms [1]. While it is a minute, flexible organelle, it is regulated by a complex array of feedback mechanisms, to the extent that small changes in MT regulation and functions can lead to devastating diseases. Consequently, MT malfunctions are implicated in a wide range of neurodegenerative diseases, including Alzheimer’s disease (AD) and Parkinson’s disease (PD) [2,3,4,5,6]. Extensive studies in recent years have focused on relating deviations in molecular and cellular functions from normal MT operation. While a great deal is already known about mutated proteins in MTs and their functions, there is a paucity of information on drug effects on MT and the regulation of mitochondrial calcium ([Ca^2+^]m) with respect to other intracellular organelles. For example, MT is connected to the endoplasmic reticulum via MAM (mitochondrial associated endoplasmic reticulum membranes), which is crucial for the regulation of [Ca^2+^] in the two compartments [7]. The rules that govern the operation of this link are not entirely clear, as are molecules that regulate MT functions.

In particular, one of the poorly studied issues is the function of MT localized in synaptic zones of dendrites, especially under dendritic spines and their immediate vicinity. MTs in these key regions are organized in relatively extended clusters, up to ≈5 µm in length. Their function may be to provide energy support to synapses and to ensure local protein synthesis in the area of clusters of dendritic spines, especially during synaptic plasticity and metabolic adaptation to stress [8]. However, the issue of signaling pathways that connect synaptic activity with MTs remains unresolved. Perhaps this role is assigned to cytosolic calcium ions ([Ca^2+^]c) [9], which may diffuse freely from the postsynaptic density to the dendritic shaft and their local MTs. In this context, the strict localization of [Ca^2+^]m in a limited area near a synapse, with very little lateral diffusion, is of particular importance. Both local calcium stores [10] and mitochondria themselves can contribute to the dynamic localization of [Ca^2+^]c.

In the present study, we employed cultured rat hippocampal neurons transfected with a molecular MT calcium sensor mtRCaMP [11] as well as other plasmids to study MT calcium regulation. Free cytosolic calcium concentration was estimated using the high affinity sensor Fluo-2. We used high resolution dynamic imaging of neuronal morphology in relation to functional changes in MT. These tools allow us to detect a correlation between local [Ca^2+^]m and [Ca^2+^]c and to propose that the latter regulates local variations in [Ca^2+^]m. In order to examine kinetics of postsynaptic mitochondrial calcium transients in response to cytosolic calcium bursts, we used two pharmacological agents, caffeine and CCCP, which are known to alter mitochondrial calcium transients. These results clarify MT functions in normal and functionally impaired neurons.

## 2. Results

### 2.1. Imaging

Two types of spontaneous events can be detected in cultured networks of neurons. The first type, the synchronized activity, which is expressed as a network burst, involves neurons generating action potentials simultaneously. During a network burst, calcium surges involve all dendrites of the neuron. These are called ‘global’ [Ca^2+^]c events. The second type of activity is ‘local’, as it is restricted to a small (2–5 μm) section of the dendrite.

The focus of our first series of experiments was on local [Ca^2+^]c events (Figure 1). Both [Ca^2+^]c and [Ca^2+^]m were imaged simultaneously at high temporal and spatial resolution (Figure 1A, see low and high power images in Appendix A), and changes in the two sensors over time could be detected (Figure 1B). Regions of interest (ROIs) containing MT clusters were traced and marked along their outer contour, whereas adjacent cytoplasmic regions were included in the ROI for surrounding cytosolic calcium measurements (see Appendix A for details). Both [Ca^2+^]c and [Ca^2+^]m local surges differed in the amplitude and duration but maintained common traits (see Appendix A). They were restricted to a small local ROI (e.g., regions in Figure 1A), typically located in close proximity to an adjacent dendritic spine. In fact, spine [Ca^2+^]c surges preceded a similar change in the parent dendrite and even more preceded [Ca^2+^]m in that dendrite (Figure 1D, top and bottom, gray dashed lines). The averaged [Ca^2+^]m response in a mitochondrion adjacent to dendritic spines had similar time course, including peak and decay time with mean duration of 246 ± 31 ms versus 295 ± 14 ms for [Ca^2+^]c detected in the dendrite (Figure 1D, top and bottom). If spine’s local calcium events are taken as 100%, the [Ca^2+^]m response could be observed in 70% of cases (along with 66% [Ca^2+^]c responsiveness in the dendrite), while nearby ‘side’ and ‘far’ MT were observed in only 18% and 5% cases with the source in the spine (Figure 1E). This indicates that local calcium events in the dendritic compartments, associated with mitochondrial clusters, typically do not spread more than 2–3 μm away from the initiation site, probably due to sequestration by the MT adjacent to the source of the event. Example spectrum images of the same region at the time of the local calcium event are shown in the middle of Figure 1B, showing that the change in fluorescence levels of both [Ca^2+^]c (top panel) and [Ca^2+^]m (bottom panel) are indeed restricted to a small region of the dendrite. Pearson’s correlation coefficient between [Ca^2+^]c and [Ca^2+^]m peaks amounted to 0.851. Furthermore, a region of the dendrite to the right of the central cluster in Figure 1A (‘F’, far) could also generate local events at a different point in time with similar correlation of 0.803, that were independent of changes in the central cluster, even though they were less than 10 μm apart (Figure 1C). Distribution of [Ca^2+^]m levels detected in ROIs adjacent to the spines and away from it (‘side’) during the [Ca^2+^]c events is summarized on Figure 1F. Statistically, the levels of [Ca^2+^]m in ROI were significantly larger than those detected in ‘side’ regions (Figure 1F,G). Moreover, the amplitude of [Ca^2+^]m transients correlated with the levels of [Ca^2+^]c. Thus, smaller cytosolic events were associated with lower [Ca^2+^]m, so that smaller transients significantly differed from medium and large ones (Figure 1H). These experiments indicate that local [Ca^2+^]m transients follow the initiating [Ca^2+^]c events and correlate with their magnitude. Hence, the small cytosolic compartments may be related to the regulation of [Ca^2+^]m levels.

We also imaged in the same cell, analyzed in Figure 1A, some global events presumably reflecting neuronal network bursts, where [Ca^2+^]c rises simultaneously in the entire dendrite (Figure 2A,B). Under these conditions, there were no differences in [Ca^2+^]c, nor in [Ca^2+^]m among three adjacent calcium compartments (Figure 2B, top panel), unlike the results shown in Figure 1B,C. Overall, a slow MT calcium rise and slow decay were seen, which began to rise with [Ca^2+^]c, and continues to be present for seconds later. Interestingly, within these slow changes, numerous fast [Ca^2+^]m transients termed ‘spikelets’ of [Ca^2+^]m, were observed against the elevated background of [Ca^2+^]m (Figure 2C). Spikelets are marked with black arrows in Figure 2B and in gray areas are enlarged in Figure 2G.

To address the functional relevance of changes in cytosolic calcium, we used two drugs known to interact with MT functions: caffeine and CCCP. Caffeine induces massive Ca^2+^ release from the ER through RyR channels, and thus promotes Ca^2+^ uptake into MTs. This release is likely to occur at MAM, which is the main site for Ca^2+^ relocation from ER to MT [2,12,13]. CCCP is a mitochondrial uncoupler, in the presence of which MTs lose their membrane potential and the ability to store calcium [14,15,16]. Effect of both drugs on global events are shown in Figure 2B, middle and bottom panels.

Moreover, in all studied mitochondrial clusters, the number of spikelets and their amplitude increased with the growth of the cytosolic response. Figure 2C summarizes the magnitude of [Ca^2+^]m surge in network bursts compared to the basal level. The contribution of the spikelets was not taken into account in the measurements, as shown in Figure 2C, gray and red areas. On the left side of the panel, one can see that caffeine not only increases basal [Ca^2+^]m levels (gray bars), but also contributed to the overall rise in network burst (red bars). Figure 2F represents the spectrum scaled net-change of [Ca^2+^]c (top) and [Ca^2+^]m (bottom) in the same area as in Figure 2A. One can note that caffeine causes a marked increase in both cytosolic and MT calcium levels. Interestingly, CCCP, unlike caffeine, did not increase either baseline or burst levels of [Ca^2+^]m (right side of panel Figure 2C). Since caffeine is known to cause Ca^2+^ release from the ER, it is suggested that spikelets reflect both the cytosolic calcium, taken up into MTs, as well as the portion, released from ER into the narrow MAM zone. Hence the frequency and amplitude of spikelets may not fully reflect the trace of a global cytosolic event.

The dynamics of [Ca^2+^]c and [Ca^2+^]m for both drugs during global calcium surge is summarized for peak (Figure 2D) and 2/3 decay timing (Figure 2E). Thus, cytosolic peak time did not differ among groups (green bars), while [Ca^2+^]m peak latency was significantly delayed in caffeine, and shortened in CCCP. When the peak timing was taken as the reference point, the 2/3 decay time was found practically the same for all groups, except for the [Ca^2+^]m in CCCP, where the decay occurred significantly faster (Figure 2E).

Caffeine caused a somewhat delayed and slow rise of [Ca^2+^]c in the dendrites. Interestingly, it the soma of neurons a much quicker, almost immediate response could be observed (see Appendix A for reference). Anyhow, following cytosolic response, an increase in frequency of global [Ca^2+^]c bursts as well as correlated [Ca^2+^]m could be seen (Figure 3A). Nevertheless, the amplitude changes in the global [Ca^2+^]c events and the rise of [Ca^2+^]m activity did not coincide in time, indicating that the latter are not a direct consequence of the calcium fluorescent changes induced by the global events. Appendix A further supports this claim by demonstrating the effect of caffeine on MT even when the cytosolic calcium indicator Fluo-2 was not used in these experiments.

Bath applied CCCP (10 μM) produced a complex action; in the first five minutes after infusion, it caused an increase in number and amplitude of global [Ca^2+^]c events (Figure 3B,E,F bottom). Next, the amplitude decreased and the events completely disappeared (Figure 3B), whereas in the case of caffeine, there was a sustained increase in the frequency and amplitude of global [Ca^2+^]c events and associated [Ca^2+^]m responses, as well as an overall increase in baseline [Ca^2+^]c and [Ca^2+^]m levels (Figure 3A,D,F top). These results confirm earlier suggestions [17,18] that an initial compartment affected by CCCP is the presynaptic terminal, causing a calcium-dependent increased release of neurotransmitters, while dendritic mitochondria are affected only sometime later. The number of global events in the presence of caffeine and CCCP at shorter and longer time scales is summarized in Figure 3E, top and bottom.

The effect of both drugs on the time course of global [Ca^2+^]c and [Ca^2+^]m is shown in Figure 3C. While caffeine increased [Ca^2+^]c amplitude, CCCP did not affect the kinetics of global events in cytosol (top). At the same time both drugs had an explicit effect on the [Ca^2+^]m timing (bottom). Thus, in CCCP (gray curve), the initial part of [Ca^2+^]m rise was faster and higher than in control (first pair of red dashed lines, marked 1), while at the later part, associated with [Ca^2+^]c decay, the caffeine-related curve (purple) was dramatically higher than both of other traces, indicating the slower decline (second pair of dashed lines, marked 2).

We hypothesize that the effect of CCCP may be associated with a gradual disintegration of mitochondrial clusters and their transformation from long filamentous structures with a large surface area to dense, rounded formations as shown before [19,20,21,22]. As a result, the entire amount of protein indicator is concentrated in a small area, creating the effect of a rapid [Ca^2+^]m rise. However, we do not have enough experimental evidence of our own to support this hypothesis. As for the effect of caffeine, the release of calcium from the ER stores is a relatively slower process, due to which the accumulation of calcium in mitochondria reaches significantly higher levels, and occurred a few seconds later.

It should be noted that the levels of [Ca^2+^]c and [Ca^2+^]m baselines for control and CCCP groups were the same but significantly elevated in caffeine (Figure 3D). The number of global calcium events was calculated at two sequential time points: at 5 and 10 min after the application of drugs compared to control. Figure 3E, top indicates the sustained increase in network activity following caffeine. In contrast, CCCP revealed a sharp increase in network burst discharges at the initial stage, followed by a dramatic drop, associated with gradual MT degradation and unhealthy state of cells.

Data on mean peak amplitude for both [Ca^2+^]c and [Ca^2+^]m are summarized in Figure 3F top (for caffeine) and bottom (for CCCP). As mentioned before, caffeine induced a sustained rise of the peak levels at both time points in cytosol (green bars) and MT (red bars). In contrast, CCCP had only short lasting and mainly insignificant effect on peak levels.

Mitochondrial transients seen on the background of a general elevation in [Ca^2+^]m during a global event in the cytosol, so called ‘spikelets’, have been analyzed and compared to local [Ca^2+^]m transients, described in Figure 1. To further analyze these temporal changes in [Ca^2+^]m relative to [Ca^2+^]c, we aligned and averaged the individual spikelets in order to compare the trace with averaged local MT calcium transients. Despite the fact that spikelets arise on the background of a general elevation in [Ca^2+^]m, their amplitude was high enough for reliable identification. In particular, we used a two standard deviation criterion for discrimination, as shown in the inset of Figure 4A. The duration and amplitude of the spikelets were similar to those of local events. Both drugs, caffeine and CCCP, have been tested in order to compare spikelets to local [Ca^2+^]m transients (Figure 4B,D). First, both drugs caused a large increase in the rate of the local spikelets, they were not synchronized (compare ‘left’, ‘middle’ and ‘far traces’ in Figure 2B). However, the amplitude and duration of spikelets and local transients in CCCP did not differ from the control, while caffeine in both cases caused a significant increase in duration [Ca^2+^]m (325 ± 23/406 ± 41 and 304 ± 28/468 ± 43) and the same applied to cytosolic local events (Figure 4B,D, black and gray vs. purple traces). Overall, CCCP produced a rather fast rise and decay of the individual events, compared to caffeine. The number of local events in caffeine also significantly increased (Figure 4E), while the number of mitochondrial transients in CCCP decreased as expected (ibid). Similarly, the number of spikelets during and right after the global event increased dramatically in control and caffeine (Figure 4G,H, periods 2–4), while in CCCP the increase was observed only at the beginning of the event (period 2), but then (period 3) a sharp decline was seen (Figure 4G,H). Most of the events disappeared in the CCCP group, compared to control and caffeine.

The similarity of the spikelets to local [Ca^2+^]m transients could be confirmed by their direct comparison in the control and in both drugs (Figure 4F). None of the difference of durations was statistically significant: 304 ± 28/324 ± 23 for controls, 468 ± 43/405 ± 41 for caffeine and 194 ± 18/171 ± 11 for CCCP.

### 2.2. Computational Modeling

To observe the temporal profile of [Ca^2+^]c in the dendrite, the total number of particles inside 1.5 μm-long dendritic partitions every 0.1 ms was counted after the initialization of the diffusion of 1000 ions simultaneous at the top of the spine head (Figure 5). It was possible to confirm that at the spine base the number of ions increased rapidly within the first few milliseconds to a maximum of 23.24 ions on average (Figure 5A, right), compared to 10.04 ions in the adjacent area. Hence, it could be concluded that any mitochondrion located on the sides of the spine base would receive slightly less than half of the ions compared to the mitochondrion located at the spine base.

Following the implementation of a spherical reflecting obstacle (Figure 5B), the rise of the number of calcium ions in the spine base was similar (23.76 on average), while the numbers in the two sides (blue and beige) were again close to half of this number (11.32 in the side that also contained a reflecting sphere and 12.04 on the other side). The conclusion was made that even under the presence of a reflecting obstacle, the amount of calcium arriving at the base of the spine were distributed approximately in even proportions into the two sides, with a delay of a few milliseconds.

It should be noted that in our model, the dendritic spine is considered as a source of calcium influx only as a physiologically plausible convention. We did not introduce specific synaptic input parameters into our model. In fact, a local source of calcium in close proximity to the MT cluster may be a form of point release from ER stores via RyR, IP_3_R or another internal source.

Then the simulation was repeated with a partially absorbing obstacle (Figure 5C) showing that the rise of the number of calcium ions in the spine base (magenta) was reduced by more than threefold (peak average = 7.08 ions), compared with the previous two cases. The [Ca^2+^] level in the empty side was now diminished by more than two-fold (2.08 ions on average), compared to this peak, while the presence of another obstacle in the side reduced the number of calcium ions to a negligible level (~0.88 ions maximum). Therefore, it is concluded that the presence of absorbing regions of calcium, similar to MT in the dendrite, could have high, non-linear reductions of calcium availability at the dendritic locations that are not directly below the spine.

## 3. Discussion

The present results demonstrate local transient changes in [Ca^2+^]m following similar changes in [Ca^2+^]c, indicating that the control of variations in [Ca^2+^]m may lie in changes in concentration of free intracellular [Ca^2+^]c. The changes are found primarily but not exclusively in close proximity to local dendritic spines, and it is likely that they are caused by activation of AMPA and NMDA receptors in the postsynaptic membrane. The difference between global and local events is likely to reflect the striking activation of voltage gated NMDA receptors caused by the strong depolarization of the dendrites [23]. Calcium-induced calcium release mechanism as well as store operated calcium entry, associated with refilling of ER stores [10,24], may also have an impact on the differentiation between network bursts and local events.

Results of the model (Figure 5C) are highly consistent with our experimental data. Thus, in the case of local cytosolic calcium events, the region of the mitochondrial cluster located near the source of calcium surge (region M, Figure 1A,B) reacts more intensely than neighboring regions of the cluster, or than a remote cluster (Figure 1C) situated 3 µm apart, where responses are not indicated at all (Figure 1E–G). In the absence of mitochondria only at one side of the focus of local calcium surge, we find asymmetric calcium distribution to the right and the left, as predicted by the model in Figure 5C. Thus, small cytosolic calcium transient can be found at the side with absent MT, while the MT-present side does not reveal any [Ca^2+^]c as well as [Ca^2+^]m signal (see Appendix A). Such uneven [Ca^2+^]c diffusion at all detected asymmetric MT clusters occurs in 79 ± 4.6% of cases.

Transient local calcium surges (dubbed ‘marks’ or ‘spikelets’) have been reported before in cultured H9 cardiac cells [12], and suggested to involve activation of ryanodine receptors. These studies employed Rhod-2, (see below) unlike our studies and their ‘marks’ have large and fast rise time and decay faster than in our cells. It is possible that this difference reflects a difference in the affinity of the sensor to calcium, or in the tissue employed for this analysis, but direct comparison has not been conducted as yet.

The frequency of spikelets increases along with [Ca^2+^]c rise. However, in control and in the presence of CCCP this increase rapidly degrades. Summarizing the caffeine-related data on duration and kinetics of both local events and spikelets, we find an increases in their amplitude, duration and a slower decay kinetics. In contrast, CCCP shortens the peak time and the frequency of [Ca^2+^]m transients. The kinetics of mitochondrial calcium decay is cell-specific. MT calcium uptake and release are important for IP3—associated cytosolic Ca^2+^ signaling [13]. The present study shows a significant kinetic change in [Ca^2+^]m release in both global mitochondrial pool and in the local areas of individual mitochondrial clusters following caffeine, which targets ER-mitochondrial interaction and following CCCP, which disrupts their physiological functionality. Differential release of Ca^2+^ from mitochondria can contribute to the control of cytosolic signaling pathways and function along with calcium stores, other organelles and internal buffers to modulate and regulate global and compartmentalized cytosolic calcium levels.

Early studies on the regulation of [Ca^2+^]m employed Rhod-2 as the calcium sensor [14,15,18]. Rhod-2 is actually a general calcium sensor, but once it enters the cell, and excess of it is washed out, it has an apparent selectivity for [Ca^2+^]m. In any case, Rhod-2 is a diffusible agent, and is not likely to predict the precise localization of changes in [Ca^2+^]m. Moreover, the introduction of Rhod2 involves an injection of the agent into the cell. The genetically encoded mtRCaMP [11,25] has several advantages over Rhod2 but has not been used much in central neurons [but, see 26]. However, the structural differences between the two agents may explain some differences in the detection, timing and duration of changes in [Ca^2+^]m.

We cannot confirm that cell-permeable Fluo-2 AM may dampen the actual measurements of MT calcium. It is unlikely for several reasons. First, the time course of [Ca^2+^]m and [Ca^2+^]c are different, and they do not share the same region of interest. Second, Fluo-2, being a diffusible sensor, may not obey the boundaries of the mitochondria, unlike mtRCaMP, and we have seen different regions of interest for the two agents (see Appendix A). Additionally, the signals recorded from mtRCaMP, when the cells were not loaded with Fluo-2, did not reveal any differences, compared to the Fluo-2-positive case (see Appendix A).

Another recent study explored the interactions between [Ca^2+^]m and [Ca^2+^]c, using both in vivo and in vitro test systems [26]. In their study, Lin et al. (2019) were able to detect a correlation between somatic [Ca^2+^]m and [Ca^2+^]c, but less so in dendritic region. Our study which focuses on synaptic region may contribute to the understanding of this difference, but further studies are needed to address the role of [Ca^2+^]m in postsynaptic functions.

CCCP is a MT uncoupler, which blocks the ability of MT to regulate cytosolic and MT calcium. CCCP has been used extensively to address issues related to mitochondrial functions. For example, Koncha et al. (2021) [16] studied the integrated response of MT to stress, using CCCP. In addition, it has been shown that albumin blocks the depolarization of MT caused by CCCP [27]. The ability of Ca^2+^ clearance of synaptic mitochondrial populations, which are more susceptible to permeability changes, was studied using CCCP [22].

Finally, a major field with special interest in MT research is Alzheimer’s disease (AD). While the main focus of recent AD studies is in the formation of amyloid plaques, there are compelling indications that malfunction of MT and intracellular calcium regulation might be the place where neurodegeneration is initiated. In line with this hypothesis, it has been found that one of the main mutated proteins associated with the disease is presenilin, localized in the mitochondria of affected individuals [28]. The role of presenilin in MT functions is currently being investigated by us and others [28]. Interestingly, among others, it is also proposed that caffeine counteracts the development of AD, because it enhances the functioning of mitochondria, which is similar to our present results [12].

In conclusion, our studies that link dendritic morphological compartments and local mitochondrial and cytosolic calcium interrelations with dendritic spines and synaptic or network activity gives new insights into dynamic mitochondrial functions within small dendritic compartments of central neurons.

## 4. Materials and Methods

### 4.1. Tissue

Animal handling was performed in accordance with the guidelines published by the Weizmann Institute’s Institutional Animal Care and Use Committee (IACUC), Rehovot, Israel, cultures were prepared as detailed elsewhere [28]. Briefly, E18 rat embryos were removed from the womb of their decapitated dames, their brains removed, the hippocampi were dissected free and placed in a chilled (4 °C), oxygenated Leibovitz L15 medium (Gibco, ThermoFisher Scientific, Inc., Waltham, MA, USA) enriched with 0.6% glucose and gentamicin (Sigma-Aldrich, Inc., St. Louis, MO, USA, 20 µg/mL). About 10^5^ cells in 1 mL medium were plated in each well of a 24 well plate. Cells were left to grow in the incubator at 37 °C, 5% CO_2_ and the medium was replaced by FUDR (5-fluoro-2′-deoxyuridine)-containing medium to block proliferating glia.

### 4.2. Plasmids and Drugs

Neurons were transfected with eGFP or eBFP, for imaging cell morphology, and MT calcium sensor (mtRCaMP), using lipofectamine 2000 (ThermoFisher Scientific, Inc., Waltham, MA, USA), at 6–7 days in vitro (DIV), and were imaged at 10–21 DIV. The transfection methodology was adopted from standard protocols [29].

### 4.3. Imaging

Cultures were initially incubated with Fluo-2AM (2 μM, Invitrogen, ThermoFisher Scientific, Inc., Waltham, MA, USA) for 1 h at room temperature to image variations in [Ca^2+^]c resulting from network activity. Cultures were then placed in the 3 mL perfusion chamber, on the stage of an upright Zeiss 880 confocal microscope using a 40× water immersion objective (1.0 NA), and imaged at a rate of 10–20 frames/s. No photobleaching was detected under these conditions. Standard medium for imaging contained (in mM) NaCl 129, KCl 4, MgCl2 1, CaCl2 2, glucose 10, and HEPES 10, pH was adjusted to 7.4 with NaOH and osmolality to 320 mOsm with sucrose. Imaging of cell morphology (405 nm), cytosolic (488 nm) and MT calcium (543 nm) were made simultaneously. All measurements were conducted with identical laser parameters for all groups (e.g., intensity, optical section, duration of exposure and spatial resolution) at room temperature. Pharmacological agents were applied into the chamber during recording. The substances were dissolved at working concentration and the initial volume was gradually replaced with this solution during 30–60 s. The substances were considered to be completely washed in or out the chamber after replacing the entire volume of the chamber two-three times.

### 4.4. Statistical Analysis

High-resolution fluorescent images were analyzed using ImageJ (NIH, Bethesda, MD, USA) and MATLAB (R2010b, MathWorks, Inc., Natick, MA, USA)-based programs. Statistical comparisons were made with post hoc tests using ANOVA and *t*-tests, as the case may be. Statistically significant differences were considered at *p* < 0.05. For data processing, we used MATLAB (R2010b, MathWorks, Inc., Natick, MA, USA), Visual Studio Code (Microsoft, Inc., Redmond, WA, USA), RStudio (Rstudio, Boston, MA, USA), KaleidaGraph (Synergy, Inc., Reading, PA, USA), and Origin (Electronic Arts, Inc., San Mateo, CA, USA) software.

### 4.5. Modeling

We simulate three-dimensional calcium diffusion in a dendritic spine and a segment of a dendrite [30] (Figure 5A). All ions are treated as independent Brownian particles in the confined geometry. The diffusion coefficient of calcium is D_Ca_ = 600 μm^2^/s^−1^. Initially (at t = 0), 1000 ions are located 10 μm below the topmost point of the spine head (top arrow). The interior surfaces of the spine head, spine neck and the dendrite act as fully-reflecting boundaries, while there are 50 calcium pumps (each with a radius of 10 nm) that are fully absorbing and uniformly distributed in the upper hemisphere of the spine head [30]. Therefore, the ions arriving at these pumps disappear instantaneously from the simulation.

The two ends of the dendrite are absorbing boundaries, separated by 5 μm, which is the length of the dendritic segment we simulated. Ca^2+^ diffusion is simulated until all ions are absorbed, either by Ca^2+^ pumps or at the two sides of the dendrite. When obstacles are added to the simulation, they are modeled as fully reflecting or partially-absorbing spheres with a radius of 700 nm, located as shown in Figure 5B,C.

All simulations were implemented with Python 3.8 and the NumPy library.

## Figures and Tables

**Figure 1 ijms-23-12321-f001:**
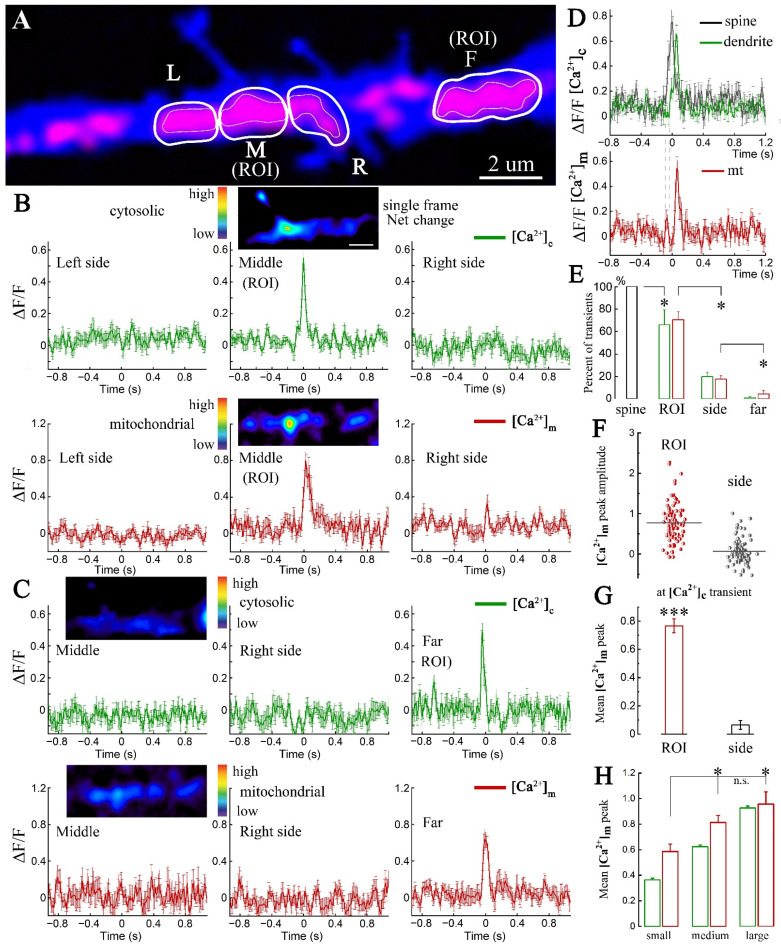
Mitochondrial calcium transients follow spontaneous local calcium events. (**A**). MT clusters (red, mtRCaMP, indicator of mitochondrial calcium ([Ca^2+^]m)) are marked by thin white boundaries, designate left (L), middle (M), right (R) and far (F) regions in the dendrite (blue, eBFP). The corresponding regions of cytosolic calcium ([Ca^2+^]c) analysis (left, middle, right and far) are marked by thicker white boundaries. (**B**) Dendritic [Ca^2+^]c (green) and [Ca^2+^]m (red) taken from panel (**A**). Averaged local calcium events in L, M (ROI), R, and the corresponding [Ca^2+^]m transients from the same regions (*n* = 10 events in one dendrite, means ± SEMs, here and in the following figures). (**C**) Averaged local calcium events in the far ROI from A and corresponding [Ca^2+^]m transients (only middle and one side regions are shown, *n* = 10). For both (**B**) and (**C**), only [Ca^2+^]c and the [Ca^2+^]m transients exceeding the threshold of two standard deviations (SD) for each trace, were included (Appendix A). (**B****,C**) Small image inserts illustrate individual (not averaged) examples of [Ca^2+^]c and [Ca^2+^]m net changes (with background subtracted) represented in spectrum palette: from blue (low) to red (high) calcium; fragment of the dendrite as in (**A**). Note the highly restricted [Ca^2+^]c and [Ca^2+^]m surge within the ‘M’ compartment in (**B**), and the lack of response in ‘M’ if the activity is targeted to ‘F’, shown in (**C**). (**D**) Top: averaged local [Ca^2+^]c events in spines (black) and adjacent dendrites (green). Note a short delay between spine and dendrite responses. Bottom: the corresponding [Ca^2+^]m (red). Two dashed vertical lines point to the initiation of [Ca^2+^]c responses in spines and dendrites, correspondingly; *n* = 14 cases, recorded from 4 segments of 4 cells in 2 different cultures. (**E**) Percentage of dendritic [Ca^2+^]c (green) and [Ca^2+^]m (red) associated with calcium surges in dendritic spines (black), taken as 100%. ‘ROI’ represent the dendritic region/MT, adjacent to the spine, ‘side’ points to the adjacent region of the same MT cluster, ‘far’ represents a different cluster nearby. Bars reflect the percentage of events either exceeding or not exceeding 2xSD criteria. [Ca^2+^]m transients: ‘ROI’ vs. ‘side’ *, *p* = 0.015; ‘side’ vs. ‘far’ *, *p* = 0.029; [Ca^2+^]c spine event vs. [Ca^2+^]c dendrite event ‘ROI’ *, *p* = 0.035, Mann–Whitney U test; percentage of *n* = 4 segments of 4 cells in 2 different cultures. (**F**,**G**) Averaged [Ca^2+^]m at dendritic ROI adjacent to spines or measured away from it, during [Ca^2+^]c surge in the initiating spine heads (*n* = 91 transients from 13 paired clusters of 6 cells in 2 cultures (*t*-test, ***, *p* < 0.001). (**H**) Ranked local dendritic [Ca^2+^]c events (green bars): ‘small’ (peak amplitude up to 0.45, *n* = 18), ‘medium’ (0.45–0.8, *n* = 55) and ‘large’ (more than 0.8, *n* = 18). Corresponding [Ca^2+^]m transients (red bars) were distributed among the same groups (data set as in ‘ROI’, D1-2,); F = 4.43674, ‘small‘ vs. ‘medium‘ and ‘small‘ vs. ‘large‘ *, *p* < 0.05; ‘medium‘ vs. ‘large‘ n.s.: not significant. One-Way ANOVA, post hoc-Fisher’s tests.

**Figure 2 ijms-23-12321-f002:**
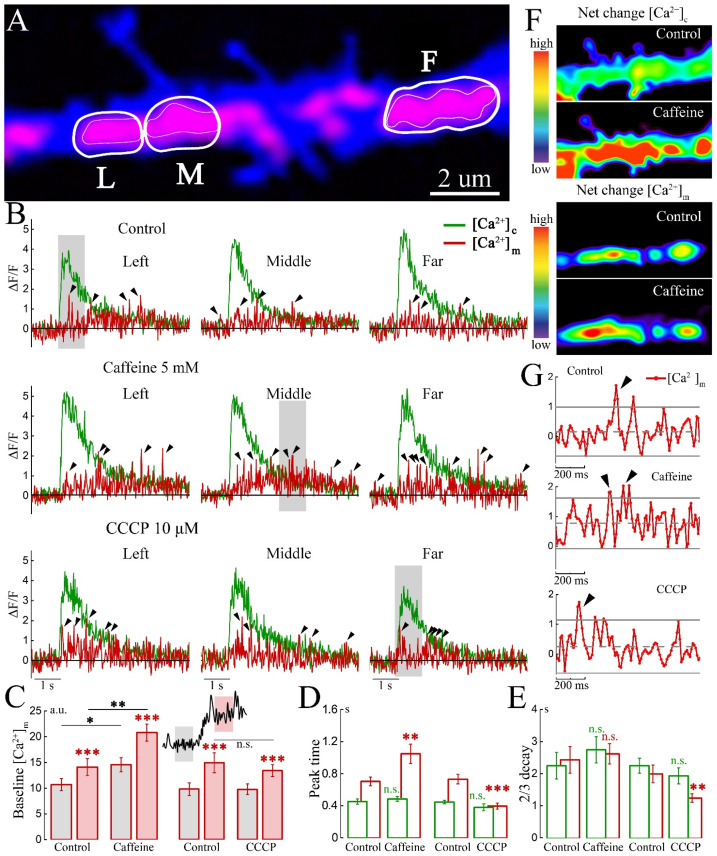
Spontaneous global cytosolic calcium events are reflected in dendritic mitochondrial calcium transients. (**A**) MT clusters (red, mtRCaMP, [Ca^2+^]m) are marked by thin white boundaries, designate left (L), middle (M), and far (F) mitochondrial regions in the dendrite (blue, eBFP). The corresponding cytosolic calcium ([Ca^2+^]c) regions (left, middle and far) are emphasized by thicker white boundaries. The dendritic segment is the same as in Figure 1A. (**B**) Top: plots of [Ca^2+^]c in dendritic regions (green) and [Ca^2+^]m (red) were made as in panel (**A**). Both, [Ca^2+^]c and [Ca^2+^]m in all regions were the same within the central cluster and in the far one. The same was seen in caffeine (middle) and CCCP (bottom) panels. Note the presence of spikelets, marked by arrowheads, during the entire global [Ca^2+^]c event, including the descending phase as well as the rise of the number and amplitude of spikelets in caffeine (5 mM), ((**B**), top and middle panels). Immediately after recording in caffeine, the drug-containing medium was washed extensively and replaced by control solution, followed by exposure to CCCP. The sequential exposure to both chemicals was only necessary to test the same area with both chemicals for illustration, but it is unlikey that caffeine had a residual effect after washout. Spikelets are marked with black arrows. (**C**) Baseline of mitochondrial calcium before and during the global calcium rise in control, caffeine and CCCP; calculated to avoid local events and spikelets, *t*-test, * *p* < 0.05, ** *p* < 0.01, n.s.: not significant. (**D**) Time from the start of the calcium rise in the global event to the maximum peak point for [Ca^2+^]c and [Ca^2+^]m in control, caffeine and CCCP. (**E**) The duration of events is calculated from the beginning of the rise in calcium to the point of 2/3 of the decay for [Ca^2+^]c and [Ca^2+^]m in control, caffeine and CCCP. (**C**–**E**) *n* = 6 cells in each comparison group, paired *t*-test, n.s.: not significant, ** *p* < 0.01, *** *p* < 0.001. An equivalent number of global events was averaged for each group. (**F**) Net change in the fluorescence of [Ca^2+^]c (top two panels) and [Ca^2+^]m (bottom two panels) in control and caffeine at the time of the global event is shown (single frames). (**G**) The 1 s scales show the [Ca^2+^]m signals during the global event in control, caffeine and CCCP containing spikelets (gray areas from panels (**B**)). For spikelets, a criterion for exceeding 2SD by at least 2 points was used.

**Figure 3 ijms-23-12321-f003:**
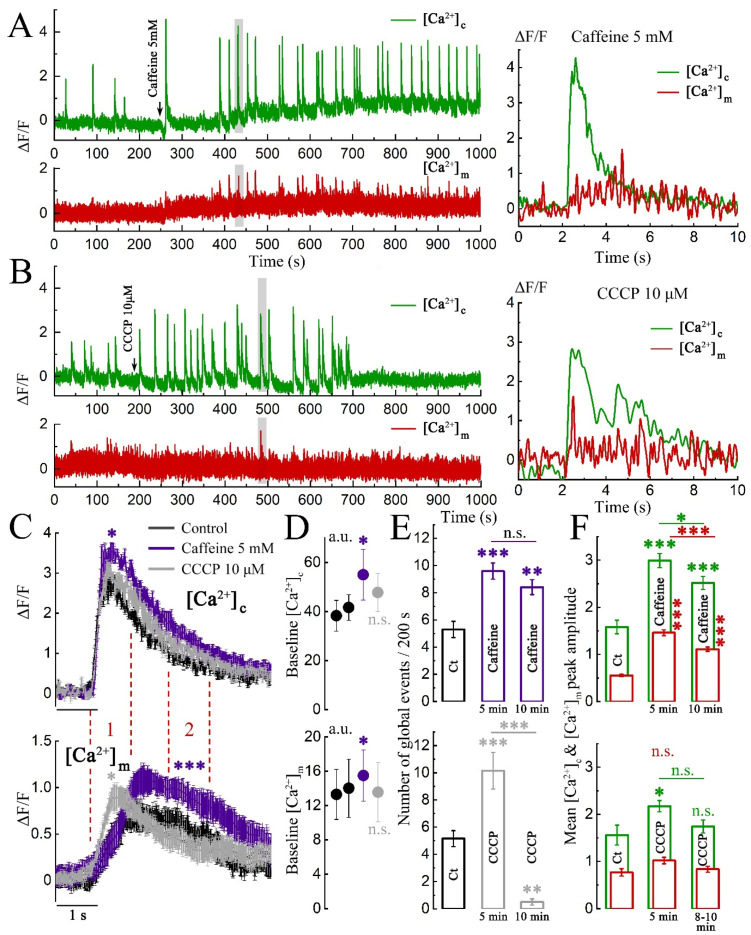
Impact of caffeine and CCCP on the kinetics of mitochondrial calcium responses associated with global events in cytosolic calcium. (**A**) Sample recording in caffeine (5 mM). Cytosolic calcium (Fluo-2) is shown in green, mitochondrial calcium (mtRCaMP) in red. (**B**) Sample recording in CCCP (10 µM). The gray areas from panels (**A**) and (**B**) are enlarged on the right, colors are the same as for (**A**). (**C**) Averages of global [Ca^2+^]c in multiple dendritic regions (top) and the corresponding [Ca^2+^]m (bottom). Control: 8 cells, 16 clusters, 2 cultures; caffeine: 4 cells, 12 clusters, 2 cultures; CCCP: 4 cells, 12 clusters. 2 cultures. An equivalent number of global events was averaged for each cluster. Normalized fluorescence at two time periods (each 1 s long) within two dashed red lines, marked 1 and 2, were compared. 1—comparison of [Ca^2+^]m during the [Ca^2+^]c peak: control vs. CCCP and caffeine vs. CCCP, *p* < 0.05, control vs. caffeine-n.s.; 2—comparison of [Ca^2+^]m during [Ca^2+^]c decay: control vs. CCCP vs. caffeine, *p* < 0.001; ANOVA, F (period 1) = 93.39, F (period 2) = 628.76, post hoc Tukey’s test. (**D**) Mean of baselines excluding global events of [Ca^2+^]c (top) and [Ca^2+^]m (bottom) in control (two overlaping black marks for each treatment), caffeine (purple), and CCCP (gray), *n* = 6 cell in each of the two comparison groups, paired samples Wilcoxon test, n.s.: not significant, * 0.01 < *p* < 0.05. (**E**) Top: number of global cytosolic calcium events in control and early (5 min) and late (10 min) caffeine. Bottom: the number of global cytosolic calcium events in control, early and late CCCP. Control vs. 5 min caffeine ***, *p* < 0.001; control vs. > 10 min caffeine **, *p* < 0.01. Control vs. 5 min CCCP ***, *p* < 0.001; control vs. >10 min CCCP **, *p* < 0.01, *t*-test, *n* is same as in (**C**). (**F**) Mean values of the peak amplitude of the global [Ca^2+^]c (green) and [Ca^2+^]m (red). Top: in control, caffeine 5 min and 10 min exposure, *n* = 6 cells, ANOVA, [Ca^2+^]c F = 17.95; [Ca^2+^]m F = 58.31, post hoc Tukey’s test, *** *p* < 0.001. Bottom: in control, CCCP 5 min and CCCP 8–10 min exposure, *n* = 7 cells, [Ca^2+^]c F = 4.60; [Ca^2+^]m F = 3.29, post hoc Tukey’s test, * *p* < 0.05, n.s.: not significant.

**Figure 4 ijms-23-12321-f004:**
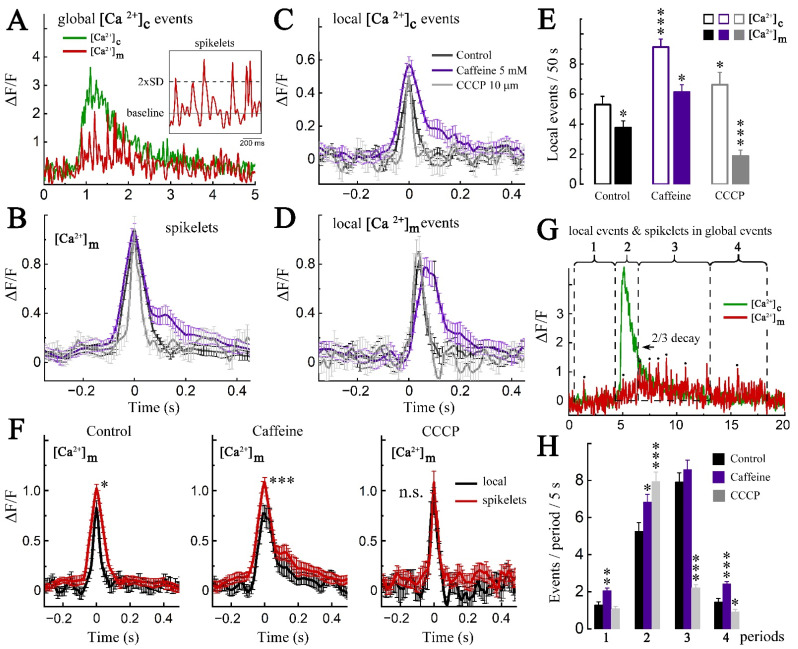
Comparison of mitochondrial calcium spikelets and local transients, recorded during global and local cytosolic events. (**A**) Example of global [Ca^2+^]c event. The calcium elevation in mitochondria during the global event [Ca^2+^]c contains separate isolated surges, «spikelets» of [Ca^2+^]m. Insert shows enlaged fragment of the recording, shown below. Only spikelets, exceeding two standard deviation (SD) criteria, were taken into account. (**B**) Averaged [Ca^2+^]m spikelets during global cytosolic elevations in control (black), caffeine (5 mM, violet) and CCCP (10 µM, gray). Multiple spikelets were collected from 4 clusters of 4 cells per each group and normalized by peak. An equal number of spikelets per cluster were averaged. (**C**,**D**) Averaged local [Ca^2+^]c transient (**C**) and corresponding [Ca^2+^]m transients (**D**), recorded from the same clusters and during the same pharmacological treatments. MT transients were synchronized with the peak in [Ca^2+^]c; *n* of averaged transients from different clusters of four cells in each group: control = 32; caffeine = 27; CCCP = 12 transients. (**E**) Number of local calcium events (empty frames) and associated mitochondrial calcium transients (filled frames) in control, caffeine, and CCCP per 50 s; *n* = 4 clusters from different cells per each group. For [Ca^2+^]c, F = 8.9924; control vs. caffeine ***, *p* < 0.001; control vs. CCCP, n.s.; caffeine vs. CCCP *, *p* < 0.05. For [Ca^2+^]m, F = 22.16832, control vs. caffeine *, *p* < 0.05; control vs. CCCP *, *p* < 0.05; caffeine vs. CCCP ***, *p* < 0.001. One-Way ANOVA, post hoc Fisher’s test. (**F**) Amplitudes of averaged spikelets in global mitochondrial calcium elevations (red) and local mitochondrial calcium transients (black) in control, caffeine and CCCP. Data set is the same as (**B** and **D**). (**G**). Sample recording of cytosolic calcium (green) and mitochondrial calcium (red), divided into four time periods: 1—before the global calcium event, 2—from its beginning to 2/3 decay of the peak, 3—to the complete cytocosolic calcium return to the baseline, 4—after the fully completed global calcium event. The numbers of local transients and spikelets were normalized to 5 s of recording. (**H**) The averaged number of local transients or spikelets per 5 s of recording in control, caffeine or CCCP according to the time periods as in (**G**). Local transients and spikelets were collected from 6 clusters of four cells per each group and their number was normalized to 5 s of recording. Period 1, F = 14.89975, control vs. caffeine **, *p* < 0.01; period 2, F = 7.32176, control vs. caffeine *, *p* < 0.05, control vs. CCCP ***, *p* < 0.001; period 3, F = 87.52905, control vs. CCCP ***, *p* < 0.001; period 4, F = 31.49251, control vs. caffeine ***, *p* < 0.001, control vs. CCCP *, *p* < 0.05; One-Way ANOVA; post hoc, Fisher’s test.

**Figure 5 ijms-23-12321-f005:**
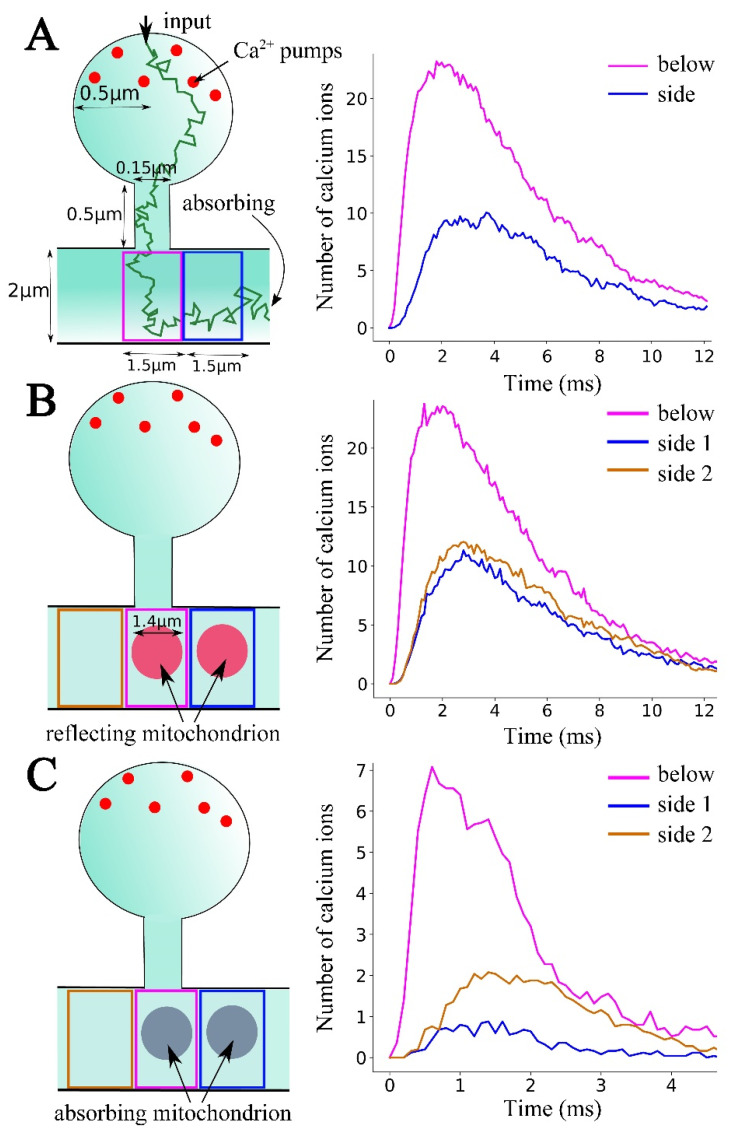
Model. (**A**) Left: Geometry of the spine model. A trajectory (green) of a typical diffusing ion starting from the spine head, until it is absorbed at a side of the dendrite. The squares represent the two cylindrical sections in which the number of calcium ions are counted. Six of the 50 calcium pumps (red) present in the spine head are also shown. Right: calcium dynamics in the dendritic section immediately below the spine (purple) and in the section next to it (magenta). (**B,C**) Left: Placement of spherical obstacles (700 nm radius), with one underneath the spine neck and the other one next to it. Each time calcium ions hit the partially absorbing obstacle (**C**), they disappear with a probability of 0.1. After the remaining 90% of the hits, the ions are reflected back to the medium. Right: Time-courses of calcium in the three areas shown in the Left panels, now under the presence of the obstacles. (Color code of beige, beige and blue also corresponds to the regions in (**A**)). All numbers of calcium ions are the averages over 25 trials.

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
