# Peer review of "Dynamic Regulation of Mitochondrial [Ca2+] in Hippocampal Neurons"

_ijms, 2022, doi:10.3390/ijms232012321_

Round 1
Reviewer 1 Report
The study by Kushnireva and colleagues describes mitochondrial calcium regulation in dendritic compartments of primary rat hippocampal neurons. In general, the manuscript is interesting, well-organized and well-written. Thoroughly prepared figures are legible and understandable. The discussion is critical and emphasizes but also explains the limitations of the research carried out. I have only minor comments regarding the manuscript:
1. Neurons isolated from distinct brain regions display unique properties and behaviors in in vitro conditions. Why did the authors choose neurons from the hippocampus and whether they did a comparative analysis with neurons from other parts of the brain.
2. Line 304: “Mt’s are complex organisms” - this term is unfortunate. Mitochondria are organelles, not organisms.
3. “Mt’s” should be changed to “Mts” or MTs thought the manuscript.
Author Response
The study by Kushnireva and colleagues describes mitochondrial calcium
regulation in dendritic compartments of primary rat hippocampal neurons. In
general, the manuscript is interesting, well-organized and wellwritten. Thoroughly prepared figures are legible and understandable. The
discussion is critical and emphasizes but also explains the limitations of the
research carried out. I have only minor comments regarding the manuscript:
1. Neurons isolated from distinct brain regions display unique properties and
behaviors in in vitro conditions. Why did the authors choose neurons from the
hippocampus and whether they did a comparative analysis with neurons from
other parts of the brain.
Response: First of all, we are very familiar with the culture of the
hippocampus, having many years of experience with it. Based on the
accumulated experience, it is easy for us to determine how "normal" the state of the neural network is and to what extent it looks "typical" for the stage of
development after cell plating. However, a much more important motivation for
us was the study of hippocampal cells, as a structure primarily suffering from
Alzheimer's type of neurodegeneration. We did not specifically study the disease model in the present study, but this is our overall goal. We assume that damage or dysfunction of mitochondria is an early manifestation of neurodegeneration, which manifests itself much later. So we decided to focus on the functions of mitochondria in a culture of hippocampal neurons, keeping in mind further study of neuropathology in this structure. Nevertheless, we consider the reviewer's proposal regarding the comparison of mitochondria in the hippocampus and other structures, for example, in the striatum or Purkinje cells, as very valuable, and we will try to apply it in further work.
2. Line 304: “Mt’s are complex organisms” - this term is unfortunate. Mitochondria are organelles, not organisms.
Response: Thank you, this typo has been corrected.
3. “Mt’s” should be changed to “Mts” or MTs thought the manuscript.
Response: Thank you, we have replaced it with MTs.
Reviewer 2 Report
The focus of the authors’ work was to monitor the intracellular calcium events in hippocampal neuron dendrites, focusing on the synchronous detection of cytosolic and mitochondrial Ca2+ levels using the mitochondrial targeted genetic Ca2+ sensor mtRCaMP and the cytosolic Ca2+ probe Fluo-2. Despite this experimental approach is interesting the strategy and drug treatment used in the work left me perplexed. The major concern is about the extensive use of CCCP as a experimental condition to monitor the physiologic response of neuronal mitochondria to cytosolic Ca2+ increase. Secondly, I found the modelling of Ca2+ ion diffusion and of the role of mitochondria in dendritic [Ca2+]c of limited usefulness for the article message. Indeed, the motivation for it and meaning out of it are not clear. The description of the rationale and a mention in the Discussion to justify the modelling are completely missing.
Despite the topic of the paper is of upmost interest for the understanding of neuronal cell biology and function and despite the described experiments are appropriately conducted, represented and explained, the overall message of the work of Kushnireva et al. is scant and the use of CCCP questionable. For these and the following reasons, I will not recommend the publication of this article on the IJMS Journal. For the sake of accuracy, I provide here a list of issues that authors may find helpful for their work.
Major issues:
- - One major concern is about the experimental strategy implemented to monitor Ca2+ dynamics in the hippocampal neurons. Sincerely I do not understand the rationale of using caffeine and CCCP. In addition, the description of caffeine and CCCP action given at lines 148-152 lacks accuracy: caffeine induces massive Ca2+ release from the ER through RyR channels, and thus promotes mito Ca2+ uptake. This likely occurs at MAM, which are the main site for Ca2+ transfer from ER to mito. For what CCCP is concerned, its common use is as a potent ionophore to depolarize the inner mito membrane thus dissipating the electrogenic force driving Ca2+ entry, thus basically abolishing mito Ca2+ uptake. Nevertheless, despite one can understand using caffeine to produce an immediate and consistent cytosolic Ca2+ surge, the implementation of CCCP to study mito Ca2+ dynamics in response to cytosolic Ca2+ transients is questionable and definitively not the most appropriate strategy. The author did not explain and justify the choice of this approach in their study. I would strongly recommend the use of a different agent to induce Ca2+ homeostasis perturbation.
- - The time scale for the local and global Ca2+ dynamics experiments (namely, Fig. 1 B1-B2 and Fig. 2 B) is not the same, it will more informative to show the traces with comparable timescale. In addition, the duration and amplitude of the cyt Ca2+ transients in local events is much reduced compared to that of global events. However, surprisingly, the peak of mito Ca2+ uptake after local events is much higher than after the sustained and consistent global cyt Ca2+ increase. This is counterintuitive and rather unexpected. Could the authors compare the two situations in terms of amplitude and duration of the cyt and mito transients and provide a possible interpretation of the differences observed?
- - The experiments with caffeine are not convincing for various reasons. Caffeine acts by inducing ER Ca2+ release from the RyR and promoting the consequent Ca2+ uptake by the mitochondria located in close proximity. Fig. 3A does not show a significant and sustained Ca2+ increase in the cytosol after caffeine addition. This is surprising and unexpected. In addition, spontaneous cyt Ca2+ oscillation were evident before drug addition and were not affected by caffeine, similar amplitude and duration of cyt Ca2+ spikes were present both before and after the addition. This suggest caffeine has little or not effect at all on hippocampal neurons. The effectiveness of the treatment should be verified.
Secondly, the mito Ca2+ increase in Fig. 2A is barely observable. Spikelets also are not evident, if they are present, the authors should at least enlarge the scale of the graph (especially the time scale) to make them visible to the reader. Indeed, the fluctuations in the mito Ca2+ signal are too close each other making it difficult to discriminate between single spike events. It would be of much help to have an X-axis scale of single sec instead of 100sec and to use arrows to indicate the intervening spikelets in the graph.
- - The authors state that “Caffeine caused an increase in frequency of global [Ca2+]c bursts as well as correlated [Ca2+]m (Fig. 3, A1)”. Sincerely, I do not see any increased frequency of cyt Ca2+ oscillations just after the addition of caffeine. If an increased frequency is observed, it occurs further after the addition, approximately after 1 min. How do the authors explain this lack of effect of the drug? Furthermore, I do not see the mentioned correlation between [Ca2+]c and mito Ca2+ elevation. As regarding to the mito Ca2+, authors should measure the amplitude of Ca2+ elevation in the three different conditions and provide quantitative data.
- - As regards to the CCCP action, also here I do not agree with the authors. It is true that the effect of the drug on [Ca2+]c is complex, but at early time points after addition the frequency of [Ca2+]c bursts seems diminished (Fig. 3 A2) and not increased as stated by the authors: ”Initially, it caused an increase in number and intensity of global events (Fig. 3, A2, & C, bottom).” Indeed, the frequency augments only after 100sec. Concerning the mito Ca2+ levels, also these data are not convincing. At lines 172-173 the authors wrote: “there was an increase in basal cytosolic calcium, followed by [Ca2+]m, both due to the high rate of bursts, as with caffeine.” I do not see any increase in global [Ca2+]m. If I considered the traces of mtRCaMP before and after CCCP, their average height is the same. Again, the authors should provide quantitative experimental data of their assertion.
- - At line 192, the authors proposed to explain the faster accumulation of Ca2+ in the mito of CCCP treated neurons by the “gradual disintegration of mitochondrial clusters”. This concept needs to be better clarified. What do the authors mean with cluster disintegration? When and why do mitochondria clusters disintegrate? Where do the authors obtain this information from? I considered most likely the option that CCCP-treated mitochondria are just permeable at ions. A transient increase in the intracellular Ca2+ levels would lead to a quicker Ca2+ entry in permeabilized mito compared to native mito with intact membranes.
- Line 201: the authors declared that “ the spikelets arise on the background of a general elevation in [Ca2+]m”, however they do not provide quantification of this elevation. Again, I would ask to corroborate any assertion with experimental data.
- - Which is the physiological relevance of these spikelets? Are they maintained in the absence of mito Ca2+ channel activity? Are they generated by the action of MCU? However, the fact that they are observed even in permeabilized mito rather suggests they are not linked to MCU channel activity…what is the authors’ proposed explanation?
- - I am wondering about the different mechanisms by which the two drugs, caffeine and CCCP, affect the spontaneous activity in the hippocampal culture: indeed, caffeine does not seem to alter the frequency and amplitude of global cyt Ca2+ events, while CCCP has a major effect on them, inducing a progressive decrease in both frequency and amplitude until the complete exhaustion of the events (Fig. 3A2). The authors did not comment, nor investigated, nor discussed the possible mechanism underlying this observation.
- - The increase in basal cytosolic Ca2+ described at lines 173-174 referring to Fig. 3 is not appreciable from the traces in the image. Authors should provide quantification of this observation and relative graphical representation.
- - Line 189-193: the CCCP addition would lead to abrogation of mitochondrial membrane potential which is the main driving force for mitochondrial Ca2+ uptake, it is then unexpected that mitochondria are still able to take up Ca2+ in this condition. How do the authors justify their observation of “a faster but decreased calcium uptake ability of Mt in CCCP “? The proposed hypothesis that “the effect of CCCP may be associated with a gradual disintegration of mitochondrial clusters leading to a faster rise time of [Ca2+]m” is not in line by the recognized role of CCCP as disruptor of the mitochondrial Ca2+ uptake driving force.
- - Is the presence of spikelets in the mt Ca2+ trace relevant in physiologic condition? Spikelets seem not to correlate, both in time and in frequency, with the intervening cytosolic Ca2+ increase, either spontaneous or drug-induced. In addition, the average Ca2+ increase in mitochondria after the different treatments and conditions is never terrific, probably not reaching the significance (anyway, authors never considered this parameter and did not provide values or statistics). Do the authors tested alternative mitochondrial Ca2+ indicators to evaluate mitochondria behaviour in the different conditions? The use of other mitochondria-targeted Ca2+ sensitive probes would help to confirm the relevance of the so-called spikelets.
General issues:
- - The meaning of the sentence in the Introduction “Both local calcium stores [10] and mitochondria themselves can contribute to the dynamic localization of [Ca2+]c” (lines 56-57) is not clear to me: do the authors intend to suggest that mitochondria act as Ca2+ buffers by locally sequestering the cytosolic Ca2+ that raised in the dendritic region of Ca2+ surge, thus preventing its diffusion? If this I s the case, it should be stated overtly.
- - In the last paragraph of the Introduction, the authors wrote that they will “clarify Mt functions in normal and functionally impaired neurons”, however, they do not use any experimental model for hippocampal neuron functional impairment. The use of caffeine as Ca2+ mobilizating agent and the use of CCCP to destroy mt membrane potential are not mimicking any of the dysfunctional condition observed in pathology. The authors should revise their sentence or clarify to what impairment they refer.
- - Fig. S. 2 puzzles me for different reasons. The signal of the mtRCaMP sensor is very noisy, especially in the inner region (the actual mitochondrial matrix, though) with fluctuations ranging far over the 50% of the whole signal (see Fig. S2). The identification of single peak events is difficult if not impossible in this condition. This behaviour is quite different to that shown in the main Fig. 1 B1 and B2. On the other hand, the entity of the global increase of mtRCaMP fluorescence reported after intracellular Ca2+ surge is limited compared to the amplitude of that reported in the cytosol. If mitochondria are the buffering agent limiting Ca2+ diffusion outside the local region of the dendrite, one would imagine they will uptake rather high amount of Ca2+. Could the authors comment and explain this lack of correlation between cytosol and mitochondria Ca2+ response? Moreover, in the main Fig. 2 the fluctuations of the mtRCaMP signal are present again, preventing to appreciate single Ca2+ events, allowing recognizing only of a global but limited Ca2+ elevation.
Minor points:
- - Line 71: the measure unit is misspelled, μm should be used.
- - At lines 82-83, the authors said that they considered the local calcium event at spines as 100% and that the [Ca2+]m response could be observed in 70% of cases. Do they take for granted that the dendritic [Ca2+]c response is also 100%? Is this the case?
- - I found the lettering of Figure panels quite unusual and uneasy, using the simple alphabetic order of letters (A,B,C, etc.) rather than introducing letter numbering (A1, A2, B1, B2, etc.) will be absolutely more convenient and definitively more appreciated.
- - The paragraph “4.2 Plasdmids and Drugs” of the Methods section do not contain description of the drugs and their use. Are they administered in perfusion or do they persist in the medium all over the experiment after addition?
- - The authors used the abbreviation Mt’s to indicate mitochondria. I find the use of the English possessive to indicate the plural of mitochondrion not appropriate.
- - Line 311: The authors wrote, “the introduction of Rhod2 involves an injection of the agent into the cell.” This is not correct. Rhod2 dye is commonly applied by incubating cells with the AM-derivative in the medium followed by several washes.
- - Line 316: The authors wrote, “A similar argument concerns the possibility that Fluo-2 AM that penetrates the cell, is also sensitive to Mt calcium concentration”. A reference for this assertion should be provided.
- - Line 340: the authors hypothesize a role of CCCP in potentiating the activity of pre-synaptic receptors. However, no investigation on the differential effect of CCCP on pre- or post- synaptic terminals has been conducted. The conclusions drawn by the authors should be calibrated in accordance to their actual results.
Author Response
Please see the attachment. (Review 2)

Round 2
Reviewer 2 Report
The authors performed extensive revision of their original manuscript; in addition they accurately gave detailed responses to all my questions providing new experimental evidence when needed.
They reshaped figures and text according to my suggestion and implemented the modifications needed to render more comprehensible and valuable their work. I am very satisfied with their revision.
Nevertheless, for the sake of precision, I would highlight few minor typos and comments before the final acceptance. The authors should proceed autonomously with their correction, after that the manuscript can be considered ready for the publication without further assessment by my side.
The authors should proceed autonomously with their correction, after that the manuscript can be considered ready for the publication without further assessment by my side.
1) Line 352 : “depolarization of the dendrites“; “of” is missing.
2) The concepts of mitochondria being subjected to multiple stressing factors and conditions, depicted at lines 385-395 is somehow far from the topic and discussion focus of the paper, which is mainly centred on dissecting Ca2+ dynamics in correspondence of physiological neuronal conditions. I would suggest to remove the paragraph, the value of the Discussion will be not affected.
3) The unit of time is missing in Suppl Fig. S4
4) Legend of Suppl Fig. S5 is unclear, please try to rephrase with all the information neede to understand the graph in a simpler way.
5) Suppl Fig. S6 is misspelled (it is named S5 in the revised manuscript).
Author Response
Dear Editors,
Thank you for the minor comments and suggestions. We have addressed all of them in the revised manuscript, as listed below. We thank you and the reviewer for your constructive comments and supportive attitude, and I am sure the comments contributed greatly to the improvement of the quality of this manuscript.
Specific responses:
1) Line 352 : “depolarization of the dendrites“; “of” is missing.Response: Added
2) The concepts of mitochondria being subjected to multiple stressing factors and conditions, depicted at lines 385-395 is somehow far from the topic and discussion focus of the paper, which is mainly centred on dissecting Ca2+ dynamics in correspondence of physiological neuronal conditions. I would suggest to remove the paragraph, the value of the Discussion will be not affected. Thank you. We removed this paragraph from the text.
3) The unit of time is missing in Suppl Fig. S4 Response: Added.
4) Legend of Suppl Fig. S5 is unclear, please try to rephrase with all the information neede to understand the graph in a simpler way. Response: You are right, the text was not clear. We re-wrote the figure caption.
5) Suppl Fig. S6 is misspelled (it is named S5 in the revised manuscript). Response: Thank you. Fixed.